# Yield reduction under climate warming varies among wheat cultivars in South Africa

Aaron M. Shew [1✉], Jesse B. Tack[2], Lawton L. Nalley[3] & Petronella Chaminuka[4]

Understanding extreme weather impacts on staple crops such as wheat is vital for creating adaptation strategies and increasing food security, especially in dryland cropping systems across Southern Africa. This study analyses heat impacts on wheat using daily weather information and a dryland wheat dataset for 71 cultivars across 17 locations in South Africa from 1998 to 2014. We estimate temperature impacts on yields in extensive regression models, finding that extreme heat drives wheat yield losses, with an additional 24 h of exposure to temperatures above 30 °C associated with a 12.5% yield reduction. Results from a uniform warming scenario of +1 °C show an average wheat yield reduction of 8.5%, which increases to 18.4% and 28.5% under +2 and +3 °C scenarios. We also find evidence of differences in heat effects across cultivars, which suggests warming impacts may be reduced through the sharing of gene pools amongst wheat breeding programs.

[1] College of Agriculture, Arkansas State University, PO Box 1080, Jonesboro, AR 72467, USA. [2] Department of Agricultural Economics, Kansas State University, 342 Waters Hall, 1603 Old Claflin Pl, Manhattan, KS 66506, USA. [3] Department of Agricultural Economics and Agribusiness, University of Arkansas, 217 Agriculture Building, Fayetteville, AR 72701, USA. [4] Agricultural Research Council, Hatfield, 1134 Park St, Pretoria 0083, South Africa. ✉email: ashew@astate.edu

Understanding the potential impacts of heat extremes on dryland wheat (*Triticum aestivum*) production is important in developing climate change adaptation strategies and recommendations for policymakers. Wheat is often produced as a supplementary, dryland staple crop to maize in Southern Africa (south of the equator), and in South Africa specifically. Wheat production in South Africa could be particularly vulnerable to extreme heat exposure given recent weather patterns[1–4], and this is only expected to continue under shifting climatic patterns for the region[4–6], creating formidable issues to maintaining agricultural production[2,7–10]. As such, wheat yield declines could present major challenges to producers and those who rely on regional wheat production for food security.

Previous studies that investigate weather impacts, and specifically heat impacts, on wheat yields have primarily been conducted in non-African settings, exist only in small experimental plots or highly controlled chamber experiments, or rely on biophysical crop models to simulate impacts where little or no in situ data exists. Empirical studies of staples such as wheat in open-air field trials are scarce over longer timeframes and for multiple locations, especially in Southern Africa[3,9,11,12]. Accordingly, this study provides important information on extreme heat impacts on in-field wheat yields using daily weather information, a multi-temporal dryland wheat dataset with observations from across South Africa, and extensive regression models. In addition, the differential impacts of warming temperatures on different cultivars are investigated and discussed with emphasis on potential avenues for producer adaptation in a warming world.

This study presents results for South African wheat production that have implications for both agricultural adaptation to climate change and food security[7–10]. Given their large domestic private and public wheat breeding programs[13–15], South African wheat cultivars may represent one of the most up-to-date germplasm to combat regional heat extremes and other production issues. Compared to many countries in Southern Africa, South Africa has a progressive agricultural sector, conducts extensive breeding efforts, and tests cultivars in open field environments at many locations across the heterogeneous growing regions of South Africa[14]. As such, dryland wheat breeding programs in South Africa provide cultivars within the country, and they distribute improved seed technologies to other African countries where both commercial and smallholder farmers may benefit[16]. Improved seed genetics can be critical for adaptation to increasing temperatures, which makes these breeding programs and cultivar-level trials particularly important[17–20].

Historically, South Africa has been the second largest wheat producer (by area and production) in Sub-Saharan Africa behind Ethiopia. In 1998 wheat acreage declined by 46% due to the deregulation of the wheat market and abolishment of the fixed pricing system by the wheat marketing board. Since 1998, South Africa has been both an importer (typically of lower quality wheat to blend with high quality domestic wheat) and an exporter, predominately to other Southern African Development Community (SADC) members. The drought of 2015–2016 saw South African wheat exports drop by over 76%[21]. The drought hit the Western Cape the hardest as it is South Africa's largest wheat growing province and is dominated (>90%) by dryland production[22]. Free State, the second largest wheat producing province, is a mix of dryland and irrigated production (where dryland wheat is the predominate method of production but irrigated wheat accounts for more than 50% of the total yield). The relatively low production of wheat across Southern Africa is principally because of abiotic (drought and heat) and biotic (Russian wheat aphid, yellow rust, stem rust, septoria and fusarium) stresses, which are increasing in intensity and frequency under climate change[14]. Previous literature has suggested that the estimated drier conditions in South Africa could reduce wheat yields from 1.8% to 4.3% annually[1,23]. Further, because of the anticipated drier conditions it is predicted that irrigation usage in South Africa will increase 6.4% a year through 2050 stressing the limited water availability even further[24,25].

Shocks to South African wheat production via heat extremes likely affect food security outcomes in South Africa and throughout Southern Africa[13,26]. Even while wheat plays a limited role in exports within Southern Africa, South African demand for wheat impacts prices throughout the region because it imports to meet demand for what it cannot produce[15,27]. Thus, when the wheat supply within South Africa decreases due to lack of production, the demand for wheat beyond South African borders increases—likely raising wheat prices regionally[28,29]. The Regional Network of Agricultural Policy Institutes (ReNAPRI) began providing wheat market outlooks as of 2015 because wheat consumption has risen for more than a decade and production has remained stagnant[27]. While maize remains the dominant staple, wheat has become important as a supplementary staple food and regardless of imports versus exports South Africa plays a strong role in mediating the quality and price of wheat within the Southern African context[15]. South African wheat has been bred for quality characteristics, and the wheat that South Africa does export typically ends up mixed with lower quality wheat to improve overall food quality in surrounding countries, including Lesotho, Eswatini, Botswana, and Zimbabwe, among others.

Some have suggested that wheat yields must increase annually by 0.86% to meet current and rising global wheat demand[9], and yields must improve despite the potentially negative consequences of increasing temperatures and changing precipitation patterns[8,30–33]. Meeting wheat demand in South Africa is made even more difficult given the extreme temperatures and drought that affect domestic agriculture throughout the country[24,26,34]. As recently as 2014–2015, approximately 22% of people in South Africa went without food[35] because of extreme drought and inadequate production of maize and wheat[21]. Such increases in food insecurity were driven by rising grain prices, which were more than 50% higher than non-drought years[35]. Other regions in Southern Africa also experienced spikes in food insecurity due to an extensive drought in 2015[21,22]. This highlights the important impacts of weather on agricultural production and subsequently on regional prices and food insecurity in Southern Africa. In addition, there is a great need to address broader climate change and food security issues in Southern Africa where 33% of people are food insecure[36,37], and climate change impacts may be more substantial than in other areas due to persistent socio-economic vulnerabilities[4,18]. Addressing these issues will require concerted efforts to understand the impacts of extreme weather and climate on agriculture, especially staple crops such as wheat, and develop strategies for creating more resilient agricultural production.

Weather and climate volatility are expected to increase in Southern Africa with both warmer temperatures and dryer conditions under global climate change[4–6]. Previous studies have focused primarily on drought impacts due to the recent events discussed above and their impacts on overall agricultural production and people in South Africa[13,34,36,38]. Although these drought events provide information on the economic and food security effects of reduced food production, there are few regional studies on the potential negative impacts of increasing temperatures on agriculture, particularly wheat. Moreover, the precipitation projections under climate change for the Southern Africa are much less certain than those for temperature increases[4,39,40], which makes them more challenging to link with

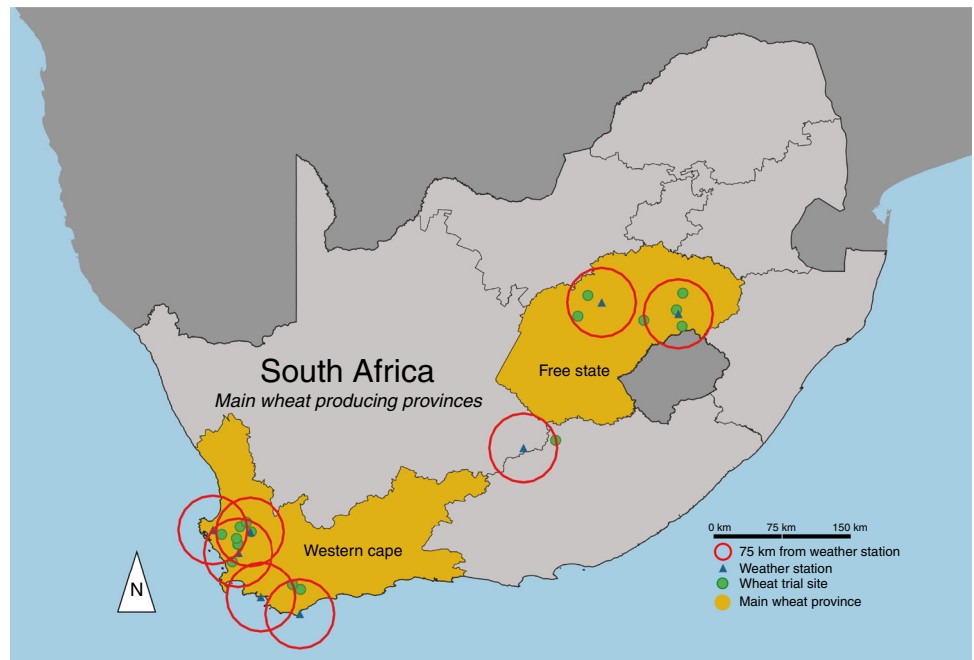

**Fig. 1 Map of the wheat trial sites and weather station locations in South Africa.** Free State and Western Cape provinces account for 73 and 99% of total and dryland wheat production, respectively, in South Africa[44]. Note, one wheat trial site and weather station is in the Eastern Cape Province, and the trial sites of Hopefield and Porterville appear as one location due to proximity. Created by A.M. Shew using QGIS 3.6.0-Noosa, under GNU General Public License v2, 1991 (gnu.org/licenses/old-licenses/gpl-2.0.en.html).

future agricultural outcomes. Global climate models simulate temperature increases between 2 °C and 6 °C in 2100 under different concentrations of $CO_2$, and the Intergovernmental Panel on Climate Change (IPCC) suggests the magnitude of land surface temperature changes could be exacerbated in South Africa with 3–4 °C by the mid-20th to late-20th century[4,40,41]. Thus, understanding the impacts of such temperature increases on wheat as a specific outcome of global and regional climate change could shed light on pathways for agricultural adaptation and resilience under future warming scenarios.

Motivated by the problems posed by warming for agriculture in the South African context and the lack of in situ studies, we concentrate our efforts on establishing the impacts of various levels of temperature exposure on wheat yields and provide important information on cultivar-level warming effects for dryland wheat production in South Africa. Our study includes a rich historical dataset with 18,881 open-air field trial observations of dryland production in 17 locations across South Africa (Fig. 1, Supplementary Table 1) and daily weather information (Supplementary Table 2). Specifically, we use a sinusoidal interpolation of daily maximum and minimum temperatures to calculate the daily time of exposure within 5 °C intervals and sum the intervals across growing seasons[42,43]. Maximum and minimum temperatures and heat exposures for the Free State and Western Cape provinces, which account for 99% of dryland wheat production in South Africa[44], can be found in Supplementary Fig. 1. Wheat yields vary substantially across the 71 cultivars, 17 years from 1998 to 2014, and locations (Supplementary Fig. 2). Wheat yield responses to heat are estimated across cultivars, years, and locations, and potential warming impacts on wheat are estimated for uniform warming scenarios spanning +1 to +3 °C following previous studies[3,11,42,43,45]. The results from these models provide key insights on extreme temperature impacts on wheat in South Africa based on expected warming for the region and outline potential adaptation strategies for climate resilience in wheat production and breeding.

## Results

**Warming impacts on wheat yields**. Temperature exposures above 30 °C are associated with large wheat yield reductions and contribute substantially to overall negative warming impacts. An additional day (24 h) of exposure to temperatures above 30 °C is associated with a 12.53% under a two-tailed test ($t(30) = 40.26$, $p = 0.000$) yield reduction on average. Parameter estimates for the preferred regression model are reported in Supplementary Table 3. The marginal effects of temperatures are provided in Fig. 2, which illustrates both beneficial and detrimental temperature exposures with similar patterns to those in previous studies[3,43]. Yield reductions from temperatures above 30 °C can be partially offset by yield increases associated with moderately warm temperatures between 25 and 29 °C. To evaluate which effect dominates, we predict yield impacts across a range of uniform temperature changes from +1 to +3 °C. All scenarios suggest warming is associated with net yield reductions ($p < 0.01$). The effects are nonlinear across uniform warming scenarios with +1 °C showing an average wheat yield reduction of 8.5% (Delta Method = −3.21, $p = 0.001$), which increases to 18.4% (Delta Method = −3.68, $p = 0.000$) and 28.5% (Delta Method = −4.16, $p = 0.000$) under +2 and +3 °C scenarios.. The average effect is a 9% per °C reduction across the warming scenarios in Fig. 3.

**Interactions between temperature and precipitation**. Importantly, our inability to control for soil moisture could bias the estimated effect of temperature exposure on wheat yields. While we cannot address this concern directly, we can include interactions between low-precipitation events during the season with our measure of heat (temperatures above 30 °C) to see if the estimated effect varies. For example, if we focus on low precipitation years—when soil moisture is likely to be a concern for wheat germination—and find the estimated coefficient on heat to be substantially different than the estimate under our preferred model, then this would suggest biased heat estimates.

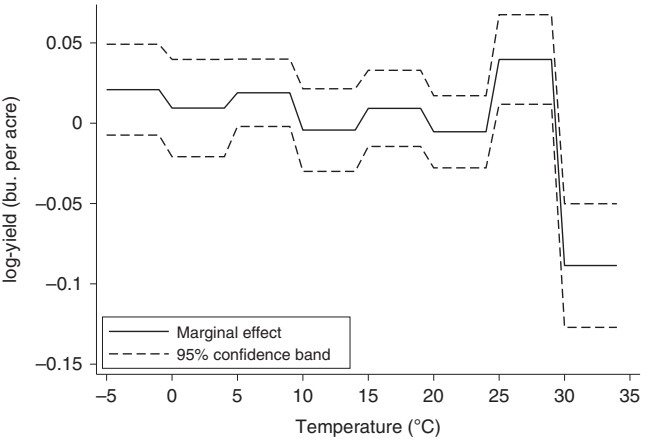

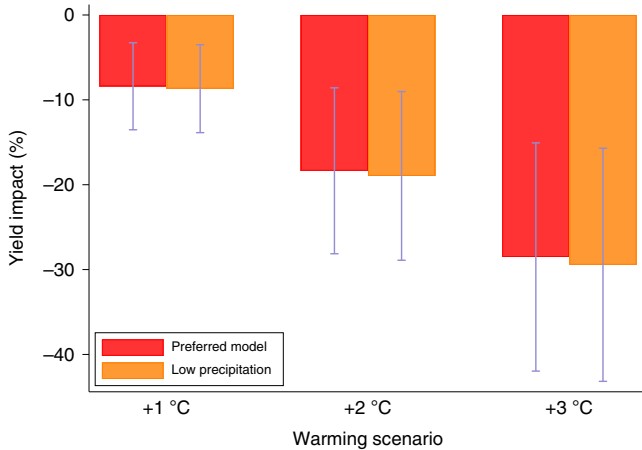

**Fig. 2 Marginal effects of temperature bins on wheat yields.** The solid line represents the change in mean log yield if the crop is exposed for one day (24 h) to each 5 °C temperature bin. Dashed lines represent the 95% confidence intervals using standard errors clustered by province-year. These results follow similar patterns to those found in Schlenker and Roberts (2009)[43] wherein yields are relatively stable across temperature bins prior to a marginal yield improvement at seemingly optimal temperatures. When critical thresholds are reached, 30 °C in this case, sharp yield reductions occur.

**Fig. 4 Warming estimates from alternative model that includes interaction of heat and precipitation variables.** Impacts are reported as the percentage change in mean yield under +1 to +3 °C warming scenarios relative to historical climate. Each 2-bar cluster shows estimates for a given scenario across different regression-model specifications. The preferred model (Supplementary Table 3) does not include an interaction between extreme heat (temperatures above 30 °C) and rainfall, while the low precipitation model does. Low precipitation is defined as a rainfall outcome below the 10th percentile of the in-sample rainfall distribution. We construct a dummy variable for this measure and then interact it with the extreme heat variable. Bars show 95% confidence intervals using standard errors clustered by province-year for $n = 18,881$ yield observations.

**More recent cultivars—higher yields, larger response to heat.** Our data include cultivars with commercial release dates spanning over two decades, and we find evidence of extensive heterogeneity across cultivars for both mean yields and the effects of heat exposure. We test genotype × environment interactions to measure the heterogeneity of heat effects across cultivars using a varying-slope multilevel model where we allow the effect of temperature exposures above 30 °C to vary across cultivars. We restrict attention here to a subgroup of 47 cultivars that have appeared in at least five trial years, and compare cultivar-specific heat effects to both mean yields (the predicted yield under average weather conditions) and the commercial release year (Supplementary Table 5). We find that more recently released cultivars have higher mean yields, with an annualized relative gain of approximately 0.7% per year; however, more recent cultivars have larger (i.e., more negative) heat effects (Fig. 5). Importantly, this tradeoff is likely benefitting producers on average as the ratio of the heat effect to mean yield is increasing over time (i.e., becoming less negative). The trends are not statistically significant ($p > 0.1$), so while the heat effect to mean yield tradeoff is insightful for potential long-term breeding it is not conclusive. We find that a similar pattern of results emerges when we focus just on the ten cultivars with the highest heat ratios from Fig. 5c (Supplementary Fig. 3).

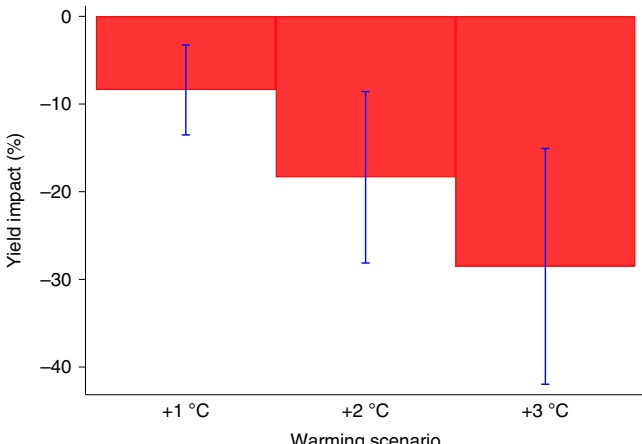

**Fig. 3 Wheat yield impacts by +1 to +3 °C warming scenarios.** Impacts are reported as the percentage change in mean yield relative to historical climate. The graph displays the warming impacts under our preferred model for uniform warming scenarios from +1 to +3 °C. Bars show 95% confidence intervals using standard errors clustered by province-year for $n = 18,881$ yield observations. Results from a uniform warming scenario of +1 °C show an average wheat yield reduction of 8.5% (Delta Method = −3.21, $p = 0.001$), which increases to 18.4% (Delta Method = −3.68, $p = 0.000$) and 28.5% (Delta Method = −4.16, $p = 0.000$) under +2 and +3 °C scenarios.

To investigate this, we allow the effect of temperatures above 30 °C to vary by interacting it with a dummy variable for precipitation outcomes below the 10th percentile of the rainfall distribution for these data. The estimates indicate a slightly higher heat effect in the low precipitation regimes as we would expect but the interaction effect is not statistically significant under a two-tailed test ($t(30) = 0.79$, $p = 0.38$), and the implied warming estimates are similar to the ones for our preferred model (Fig. 4).

**Potential adaptation through breeding.** The heat effect parameters (coefficients on 30 °C+ bins) exhibit a wide range of heterogeneity across cultivars with the highest estimate approximately twice as large as the lowest (Fig. 5), which suggests potential opportunities for adaption via selective breeding and optimal cultivar selection. We use these estimates to simulate a switch from the cultivar with the highest effect to the lowest, quantified by the change in yield impacts for the alternative warming scenarios. Results suggest that mild (~1 °C) warming impacts are approximately 50% smaller for the more resilient cultivar and significant reductions can still be achieved at higher

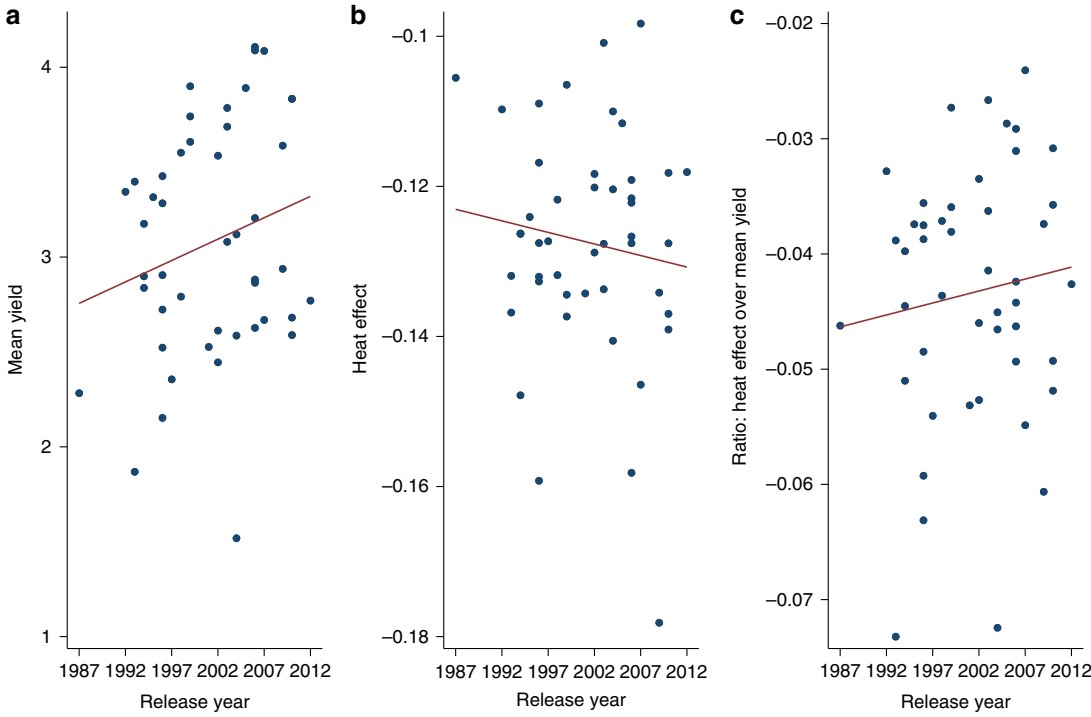

**Fig. 5 Mean yields and the effect of heat across wheat cultivar release years.** The figure shows the tradeoff between mean yields and the effect of temperature occurrences above 30 °C (heat effect) across cultivars based on the year they were commercially released. Data points are for specific cultivars and lines are linear trends. Panels **a**, **b**, and **c** report values for mean yields, heat effects, and the heat effect normalized by mean yield, respectively.

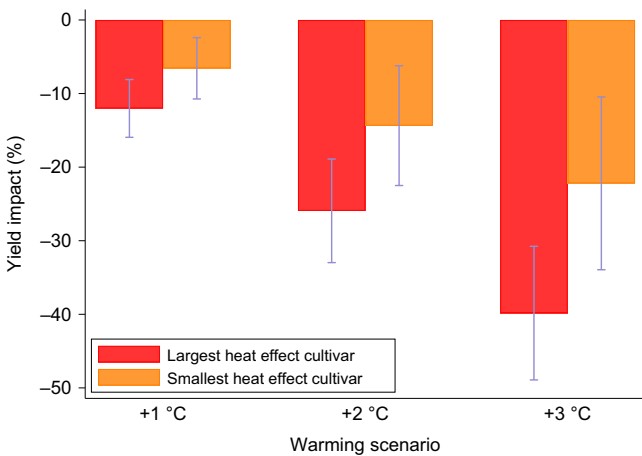

**Fig. 6 Comparison of warming impacts on wheat yields for the cultivars with the largest and smallest heat effects.** We allow the effect of temperature exposures above 30 °C to vary across cultivars. Impacts are reported as the percentage change in mean yield under +1 to +3 °C warming scenarios relative to historical climate. Each 2-bar cluster shows warming impacts for the cultivars with the largest and smallest heat effect. Bars show 95% confidence intervals using standard errors clustered by province-year for $n = 18,881$ yield observations. .

warming scenarios. The impact reductions across the 1 to 3 °C warming scenarios are 5.5, 11.5, and 17.6 percentage points, respectively (Fig. 6).

Cultivar-specific differences could play a meaningful role in helping policy-makers, wheat breeders, and agronomists further support and develop more weather-resilient wheat in the face of climate change. Accordingly, we found that recently introduced cultivars have higher average yields but larger extreme heat effects

(Supplementary Table 5), which highlights a potential avenue for breeding efforts in South Africa targeting warming impact reductions. The five cultivars with the lowest heat effects above 30 °C (with release year in parentheses) were PAN3118 (2003), PAN3144 (2007), SST124 (1987), SST399 (1999), and Tugela-Dn (1992) with impacts between approximately −10 and −11 percent relative to the multi-variety average, while the highest heat effects occurred for cultivars PAN3368 (2009), SST367 (1996), SST356 (2006), Limpopo (1994), and PAN3355 (2007) with impacts between approximately −14.5% and −17%. Importantly, all of these cultivars represent breeding efforts from the three largest independent wheat breeders in South Africa. These results enhance our understanding of how extreme temperatures will impact dryland wheat yields under current conditions and for future climate change scenarios, as well as the potential for adaptation through breeding efforts.

## Discussion
In recent years, the publically funded South African Agricultural Research Council-Small Grains Institute (ARC-SGI) and the two private South African wheat breeding programs (Pannar and Sensako) have pursued improved pest resistance, grain yield, and grain quality, e.g., milling and end-user standards[15,46], which may explain the yield gain tradeoff for higher heat impacts on yield. Historically, South African wheat breeders have been most constrained by the strict quality (rheological and baking characteristics) requirements needed to commercially release a cultivar. Due to the complex genetic properties of wheat and its characteristic self-pollination[47,48], other factors such as heat tolerance or water-use efficiency have potentially been overlooked as less significant for meeting producers' needs when compared to wheat yield gains and meeting strict quality standards[13]. It is also possible that heat stress tolerance is less attainable because of inadequate genetic information for these types of wheat traits in

South Africa[49,50]. We find evidence of substantial differences in heat effects across cultivars, which suggests the prospect of reducing warming impacts through the potential sharing of gene pools amongst existing wheat breeding programs, whether public or private.

In Africa more generally, understanding the potential impacts of warming temperatures and extreme weather on staple foods such as wheat is critical for meeting current and future food-security needs[9] and has important implications for global climate change, agrarian adaptation, plant breeding, and agricultural policy[18,19,24]. Southern Africa will likely face tremendous challenges associated with global climate change, even if major changes restrict warming to +2 °C. Given the findings in this study, wheat breeding programs in South Africa may focus on combining heat tolerant cultivars with the more recent higher yielding cultivars in an attempt to provide wheat producers an avenue to reduce the effects of global climate change. Moreover, increased adoption of irrigated wheat could potentially help maintain yields under warming scenarios in Southern Africa[34], but further research is warranted on this topic. Increasing temperatures will require adaptation among many sectors of society globally, but agriculture in rainfed Africa, and wheat production specifically, could experience disproportionately more negative effects of a warming climate than other regions[8,10,11,51,52], which makes the results from this study important. High levels of vulnerability, food insecurity, and a lack of access to improved seed technologies with targeted heat resilience may inhibit the ability of agrarian communities to adapt to weather volatility[18,19,24,53,54].

A lack of empirical data for agriculture in many food insecure countries, and in Southern Africa specifically, has led to the predominance of biophysical crop model simulations to derive climate change impacts[1,2,8,55]. Empirical studies such as this may provide important and improved information for synergistic analyses in biophysical crop models. In a meta-analysis, one study[8] found mean yield changes for wheat under climate change to be –17% in Southern Africa by 2050. Importantly, these findings, based primarily on projected and simulated crop outcomes[23], are reinforced by the findings in this study. Our results more specifically suggest that wheat yields in South Africa may not be as susceptible to increased temperatures and global climate change as those in the broader Southern Africa context. Future research may pair these results with those of others[2,8,11,20,33,56] to investigate opportunities to decrease heat impacts on wheat yields in other regions of Southern Africa by supporting cultivar development and distribution from South Africa throughout Southern Africa. The results from this study could be synergistic with biophysical crop growth models in examining global climate change adaptation potential. Importantly, there are tradeoffs between empirical estimations as in this study and biophysical crop models. For an extensive discussion of this topic, see Lobell and Asseng (2017)[55] and Roberts et al. (2017)[12].

Research on behavioral adaptation (e.g., optimal cultivar selection) and technological innovation (e.g., targeted breeding efforts) is critical for reducing warming and drought impacts on wheat both at present and under global climate change[57,58]. Our study was limited to wheat cultivars and weather patterns commonly found in South Africa and omits $CO_2$, which may be important for understanding tradeoffs in long-term climate change impacts on biomass and yield. To our knowledge, the in-field wheat dataset coalesced from regional sources and presented herein is the largest in Southern Africa, but the results may not be representative of other regions in Southern Africa depending on where those cultivars have been developed. Collecting empirical data across Southern Africa is particularly challenging given the lack of repeated in-field measurements at similar locations and through time, and without such repeated measurements it is difficult to obtain the required in-sample variation necessary for the extensive regression models employed in this study. Moreover, this study focused primarily on dryland wheat production, but in fact, many farms in Southern Africa have some irrigation capacity. Future work should investigate irrigation offsets in production and explore potential adaptation strategies.

In summary, this work provides results for heat exposure impacts on dryland wheat production in South Africa. Findings suggest that large reductions in wheat yields occur when temperatures exceed 30 °C and that warming impacts will increase non-linearly under uniform warming scenarios. Our results indicate that concentrated efforts should be made to integrate traits that reduce heat effects into more recent cultivars to maintain and improve opportunities to offset warming impacts. This is of utmost importance if food security needs are to be met in South Africa and the entire Southern Africa region. As scientists, policy-makers, and agrarian communities strive to address food insecurity, climate change impact information as provided in this study could play a pivotal role in how adaptation strategies are created and supported at the cultivar level.

## Methods

**Experimental design.** Raw data used in the manuscript were collected by the ARC-SGI of South Africa in open field test plots. The raw data include observed dryland wheat yields matched by location with daily minimum and maximum temperatures and total precipitation recorded during the growing season at near-by weather stations. Weather station data were downloaded from NASA GSOD using GSODR[59]. From the raw data, we only include wheat trial locations that have a weather station within 75 km and at least five years of wheat field trials, and wheat cultivars must appear in at least two trial years. It is possible that there are slight differences in the weather station observations and the actual weather at wheat trial locations, particularly with respect to precipitation. However, the weather station observations in the study region appear representative based on climatic norms[60] and are the best available data for capturing daily extremes. This results in 18,881 wheat yield observations from ARC-SGI spanning 17 locations and 71 cultivars from 1998 to 2014.

The yield and weather data vary substantially in-sample, which supports robust estimation of wheat yield responses to extreme and average weather conditions (Supplementary Tables 1, 2; Supplementary Figs. 1 and 2). The growing season for each location-year-cultivar is defined by the planting and harvest dates, and typically span mid-May to late October. Planting and flowering dates are observed, not estimated. Flowering is defined as the day at which 50 percent flowering occurs. Harvest dates are not observed and thus inferred using a rule of 30 days after the observed flowering date, which provided consistent results with other alternatives discussed in the robustness checks below. Phenological information were collected by both ARC staff and wheat producers based on a field observation conducted once per weekday for ARC run stations and daily for producer fields. The temperature bins are calculated from maximum and minimum temperatures using a sinusoidal interpolation of temperature exposure within each day and span 5 °C intervals. Total days (24 h) spent within intervals for the entire season are summed into eight temperature exposure bins. All negative temperatures are summed into a single bin, as well as all temperatures above 30 °C. Notably, exposures greater than 30 °C occur substantially more in the Free State compared to the Western Cape.

**Statistical analysis.** The preferred regression model specifies log wheat yield as a function of location, cultivar, and year fixed effects, as well as a quadratic polynomial for cumulative precipitation and the eight temperature bins mentioned above[61]. The weather variables are seasonal aggregates from the observed planting date to the inferred harvest date. The highest temperature bin of >30 °C represents exposures known to negatively affect wheat yields[62–64]. Average exposures across bins are provided for the two main dryland wheat-growing provinces, the Free State and Western Cape, in Supplementary Fig. 1. We considered simplified models that include linear and quadratic trends instead of year fixed effects, or (alternatively) omitting the temperature bins in favor of average temperatures, and found that they substantially reduced model performance (Supplementary Tables 3, 4). In addition, we considered extensions of the model that added pre-season precipitation (30 days before planting), or (alternatively) a cubic polynomial for in-season precipitation instead of a quadratic, and found that they also did not improve model performance. The preferred model is specified in Eq. (1):

$$y_{ijt} = \alpha_i + \alpha_j + \alpha_t + \beta_1 p_{ijt} + \beta_2 p_{ijt}^2 + \sum_{k=1}^{8} \delta_k Bin_{ijkt} + \varepsilon_{ijt}, \qquad (1)$$

where $y_{ijt}$ is log yield for cultivar $i$ in location $j$ in year $t$. Fixed effects ($\alpha$) are included separately for cultivars, locations, and years. The weather variables

**Table 1 Moran's I (MI) spatial autocorrelation for log yield and regression errors.**

| Data | MI-1 km | MI-100 km | MI-500 km | MI-1000 km |
|---|---|---|---|---|
| Log yield | 0.34404 | 0.319946 | 0.157958 | 0.097185 |
| Regression errors | 0.178149 | 0.138631 | 0.058130 | 0.026231 |

include a quadratic polynomial effect for cumulative precipitation $p_{ijt}$ and the nonlinear effect of weather across temperature bins $Bin_{ijkt}$.

There is likely a large amount of spatial correlation among the error terms of the model across cultivars in the same location, as well as across locations more generally. One could cluster standard errors by year to account for all spatial correlations, however there are 17 years in the data which is a questionably small number of clusters[65,66]. Instead we cluster errors by year-province as there are only two provinces in the data, Western Cape and the Free State, and their boundaries are several hundred kilometers apart. This method accounts for correlations among the regressors which can also bias standard errors. Cameron and Miller (2015)[65] report the variance inflation factor in their equation 6 as $1 + \rho x \, \rho u \, (N-1)$, where $N$ is the cluster size, $\rho u$ is the within-cluster correlation of the regression errors, and $\rho x$ is the within-cluster correlation of the regressor. Note that spatial correlation of the regressors can bias regression standard errors downward even if the errors are only slightly correlated. Just under their equation 6, Cameron and Miller (2015)[65] cite a study in which the correlation of the errors was small at 0.03 but the inflation factor was 13 because the regressors were highly correlated.

To better investigate the role that spatial correlation is playing in this analysis, Moran's I was calculated for each year in the data for both the log yield observations and the regression errors from the preferred model above. The averages of the Moran's I across years is presented in Table 1. The distance of 1 km captures the within-trial correlations, whereas the distances 100, 500, and 1000 km capture broader groupings. Positive correlation exists in the log yield data and it is highest within-trial as expected. The correlation remains positive as distance increases but dilutes to its smallest value at 1000 km. The regression purges much of the correlation from the data as indicated by the Moran's I for the errors, although some remains. As noted above, the clustered errors may still produce an adjustment by increasing the variances compared with classical Ordinary Least Squares which does not account for correlations. For example, the standard error on our measure of heat (the >30 °C bin) is 0.0196 under clustering but 0.00433 without clustering (i.e., robust standard errors). This suggests an inflation factor of approximately 20 which is quite large and important for adequately representing the statistical uncertainty in our warming impacts.

Heterogeneous cultivar-level temperature effects are investigated to assess the potential for climate change adaptation via cultivar selection. The preferred specification was modified to account for differences in cultivar effects using the following multilevel model[66] specified in Eq. (2):

$$y_{ijt} = \alpha_i + \alpha_j + \alpha_t + \beta_1 p_{ijt} + \beta_2 p_{ijt}^2 + \sum_{k=1}^{8} \delta_k Bin_{ijkt} + u_i Bin_{ij8t} + \varepsilon_{ijt}, \quad (2)$$

where we extend the preferred model to include a random slope ($u_i$) across cultivars for the highest temperature bin (30 °C+). Note that the fixed effects from the preferred model are include here as dummy variables in the fixed portion of the multilevel model. The only random effect in the multilevel model is for the effect of the >30 °C bin.

Warming impacts are based on uniform changes in the daily temperature data. For example, we use the observed (historical) daily minimum and maximum temperatures and increase them by 1 °C and then re-calculate the growing season bins for all locations and years[3,43,45]. Averaging these across years and locations then provides a shifted climate to simulate yield change based on the initial regression model parameters and yield estimates. The impacts are calculated as $100 \left[ e^{(Bin\,1 - Bin\,0)\delta} - 1 \right]$ where **Bin** is a vector of the temperature bins for shifted (1) and baseline (0) climate. The same steps are repeated for the 2 and 3 °C warming scenarios as well. Estimates from the regression in Supplementary Table 3 are used for $\delta$. The point estimation for warming scenarios relies on the Delta Method of asymptotic approximation for large samples as implemented via the nlcom command in Stata version 16.

**Robustness checks**. The first robustness check we consider is replacing the temperature bins with a quadratic specification of seasonal average temperatures. Interestingly, a two-tailed joint test under this model implies that temperatures do not have a statistically significant effect on yields ($F_{(2,30)} = 0.57$, $p = 0.5716$), thereby suggesting that seasonal averages cannot capture yield reductions associated with heat above 30 °C as in our preferred model. The seasonal average model generates misleadingly small warming impacts (Supplementary Fig. 4).

Next we investigate the appropriateness of the equally spaced five degree exposure bins by examining three alternatives: (i) bins of length three degrees, (ii) bins of length five degrees but with a threshold of 29 °C and, separately, (iii) a threshold of 31 °C. We find that all three alternatives produce similar marginal

effects of temperatures and warming impacts as our preferred model (Supplementary Figs. 5 and 6).

Under our preferred model the parameters for precipitation and precipitation squared are statistically significant for a two-tailed joint test ($F_{(2,30)} = 9.43$, $p = 0.0007$). We find that the yield effects of precipitation are not trivial as a one standard deviation reduction in cumulative rainfall below the average level is associated with a 9.6% yield reduction. To more directly investigate the differentiated impacts of drought and heat, the precipitation component was modified to include the quadratic function (as in the preferred model) along with an indicator variable that takes on a value of "1" when cumulative precipitation is below the 10th percentile of all observed rainfall data. This indicator captures low rainfall conditions likely associated with droughts, and findings suggest the effect of 10th percentile rainfall is an 18% yield reduction (Delta Method = $-2.95$, $p = 0.003$). The inclusion of the additional low-rainfall control variable produced similar marginal effects of temperatures and warming impacts as our preferred model (Supplementary Figs. 7 and 8). In addition, we consider controlling for the seasonal variation of precipitation as in Rowhani et al. (2011)[67], but found a similar pattern of results for the temperature and warming effects (Supplementary Figs. 8 and 9). Thus, the high temperature effect and precipitation effect seem well differentiated from each other, likely due to the location and year fixed effects that control for (among other things) locations with a more drought-prone climate and widespread droughts across locations within years.

It is essential that cultivars in the data experience sufficient heat exposure to capture the temperature effects, especially when we estimate the cultivar-specific heat effects. Within the sample, every cultivar was exposed to temperatures above 30 °C ranging from 4 to 115 h. Not every cultivar was exposed to temperatures above 30 °C at every location, but cultivars with no exposure above 30 °C at every location account for less than 10 percent of observations. Nonetheless, as a robustness check for the warming impact estimates we drop cultivar-years not experiencing exposures above 30 °C and re-estimate the model. We find similar marginal effects of temperatures and warming impacts as our preferred model (Supplementary Figs. 9 and 10).

We also consider whether allowing the temperature and precipitation effects to vary within season affects the warming impacts. We separate the growing season into three stages: (i) planting to 20 days before flowering to capture the vegetative stage, (ii) 20 days before to 10 days after the flowering date to capture the flowering stage, and (iii) 10 days after flowering to the end of season to capture the grain-filling stage. We then re-estimate the model including stage-specific measures of the precipitation and temperature variables, and find that warming impacts are very similar to those from our preferred model approach (Supplementary Fig. 11).

Next, we analyze whether cultivars developed from specific breeders provide differential heat effects by interacting the temperature bin variable for exposures above 30 °C with dummy variables for each of the three breeders represented in our data: Pannar, Sensako, and the South African ARC-SGI. A two-sided joint test of these interactions suggest that the heat effects do differ across breeders for $n = 18,629$ yield observations with breeder information ($F_{(2,30)} = 6.68$, $p = 0.004$), however the magnitude of the differences are small and the warming impacts are similar across all three breeders (Supplementary Figs. 12 and 13). We also consider whether heat effects differ across the spring, facultative, and winter wheat cultivars represented in the data using the same dummy variable approach. A two-sided joint test suggests a lack of statistical significance for these differences ($F_{(2,30)} = 2.20$, $p = 0.128$), and the temperature and warming effects are similar across all three types (Supplementary Figs. 13 and 14).

Another robustness check interacts the temperature bin variable for exposures above 30 °C with a continuous variable for the year that each cultivar was publicly released. The in-sample release years span 1984–2012 and we again find a lack of statistical significance for the interaction with a two-tailed test ($t_{(30)} = 0.53$, $p = 0.471$) coupled with similar temperature and warming effects (Supplementary Figs. 13 and 15).

The robustness of weather station data was tested by including all available weather stations within 200 km (regardless of missing data) for every wheat trial location using distance-weighting ($1/\text{distance}^2$) of the weather observations at the location-year-day level. This increased the number of field trial sites to 32 (some were dropped before because of missing weather data) and the number of unique weather stations to 107. The number of stations matched to a particular site ranged from 12 to 30. We then re-estimate the model using these alternative data and find that the temperature and warming effects are similar to the preferred model (Supplementary Figs. 16 and 17). It is also possible that this distance-weighted interpolation approach is overly simplistic, thereby introducing measurement error that can bias estimates. This type of error would likely affect precipitation more than temperature due to its more localized nature, so we replace our measure of rainfall with that of the gridded Climate Hazards group Infrared Precipitation with Stations (CHIRPS) dataset[68]. We re-estimate the model and find that the temperature and warming effects are again similar to the preferred model (Supplementary Figs. 16 and 17).

Some studies have shown that wheat maturity occurs more quickly under heat stress[62,69]. Thus, to test our assumption of a flowering-to-harvest time of 30 days at each location-year, we use this expanded weather station data and re-calculate the temperature bins for a shorter 20 day maturity period. We define the optimal maturity length by running separate regressions of log yield on the weather covariates for each location-year in the data. Each iteration produces two measures

of R-squared, one for each of the two maturity lengths, and the higher one is used for that location-year. We find that 30 days is optimal for approximately 2/3 of the location-years (Supplementary Fig. 18). A regression of the improvement in R-squared from varying the maturity length on the occurrence of temperatures above 30 °C suggests that a one percent increase in heat occurrence only improves model fit by approximately 0.001 percent. In addition, we find that optimizing the maturity lengths by location-year produces similar temperature and warming effects as the preferred model (Supplementary Figs. 16 and 17).

Expanding the weather data also provides an opportunity to consider the potential effects of shifting planting dates. Producers may adapt to increasing heat stress by planting earlier to avoid critical periods of heat exposure. To test the implications of this adaptation, warming impacts were simulated based on the initial temperature impacts with different weather variables created by planting date shifts at 7 and 14 days earlier with fixed (days-to-flowering and days-to-harvest) season lengths. For +1 °C, shifting planting dates to 14 days earlier provides approximately one percent reduction in the warming impact on yields, while for +3 °C a 14 day earlier planting date may reduce impacts by about four percent (Supplementary Fig. 19).

**Reporting summary**. Further information on research design is available in the Nature Research Reporting Summary linked to this article.

## Data availability
The datasets generated during and/or analyzed during this study are available in the Harvard Dataverse repository, https://doi.org/10.7910/DVN/8Y6Q7F.

## Code availability
The codes used for statistical analysis in this study are available in the wheatcultivars_heat_zaf repository on Github, https://github.com/amshew/wheatcultivars_heat_zaf.

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

## Acknowledgements

This work was supported in part by the National Science Foundation (NSF) Graduate Research Fellowship Program [Grant No. DGE-1450079]. The authors also acknowledge the South Africa Agricultural Research Council-Small Grains Institute (ARC-SGI) for providing access to the data used in this study. All opinions expressed in this paper are the authors' and do not necessarily reflect the policies and views of NSF or ARC-SGI.

## Author contributions

A.M.S., J.B.T., L.L.N., and P.C. conceptualized the study and performed research. P.C. collected and synthesized data. A.M.S., J.B.T., and L.L.N. designed the study. A.M.S. and J.B.T. analyzed data. A.M.S., J.B.T., and L.L.N. wrote the manuscript. A.M.S., J.B.T., L.L.N., and P.C. revised the final manuscript.

## Competing interests

The authors declare no competing interests.
