## [Peer Review File · Nature Communications]

Reviewers' Comments:

Reviewer #1:

Remarks to the Author:

General comments:

The submitted manuscript presents a statistical analysis on the high temperature effect on wheat yields in South Africa. The variety-specific data examined here are potentially novel and useful to address the scientific question considered here and lead to implications for breeding targets. The manuscript is mostly clearly written. However, I have several concerns from the methodological point of view, as listed below. Given the current manuscript, rationales are not sufficient or in part lacking to conclude that this study is technically sound; and the conclusions are supported by evidence.

Major concerns:

1. My main concern is that whether only limited samples in terms of varieties and locations are used to estimate the yield sensitivity to the temperature bin $>30\text{degC}$, although the total sample size is quite large (18,881). Figures S1 and Table S1 suggested that wheat grown in Western Cape is rarely exposed to temperatures $>30\text{degC}$. In Free State, the exposure to the high temperatures may happen for some varieties but not for all varieties as many varieties complete their crop duration by up to the end of September in which the high temperatures rarely occur. The incidences of temperatures $>30\text{degC}$ are shown in Fig. S1, but it is likely that these data came from limited cultivars and locations and probably the sample size is very small. If this is the case, the estimates of yield sensitivity to the high temperatures do not represent the average of various varieties examined here. Subsequently, the comparison across the varieties with low and high sensitivities presented becomes meaningless in this case. The author(s) is strongly encouraged to provide rationale that their estimates on the temperature bin $>30\text{degC}$ is derived based on a wide-ranging varieties and locations.
2. I am not convinced how the findings of this study have implications in particular for Sub-Saharan Africa (SSA), whereas the author(s) sometimes emphasize it in main text (for instance, "... as a major wheat producer and consumer, shocks to South African wheat production via heat extremes likely affect food security outcomes in South Africa and throughout the southern SSA region" (line 48-50)). In reality, South Africa is not necessarily a major wheat producer in the world. According to wheat production and trade data in 2017 available in FAO statistical database, South African wheat production is ranked 5th among African countries following to North and West African countries (e.g., Egypt, Morocco, Ethiopia and Algeria). In 2017, South Africa produced wheat of 1.5 million tons (Mt) and imported 1.7 Mt mainly from Russia and Germany (Table 8 in https://apps.fas.usda.gov/newgainapi/api/report/downloadreportbyfilename?filename=Grain%20and%20Feed%20Annual_Pretoria_South%20Africa%20-%20Republic%20of_3-25-2019.pdf). Wheat exports from South Africa were relatively small (0.08 Mt). Given this situation, the present study may have implications for domestic production in South Africa, but possible implications for the remaining region in SSA as well as those for international scientific communities are unclear.
3. Sowing date shifts are widely considered as an adaptation measure in crop production. However, this study assumes that the crop durations are unchanged in a warmer climate (+1 to +3 degC). This assumption is unreasonable and unrealistic, and more importantly leads to an overestimation in yield impacts associated with the high temperatures. Therefore, the yield impacts estimates presented need be revised by incorporating adaptation measures that would occur thanks to less additional costs for producers.
4. Although I am not sure specific targets in wheat breeding in South Africa, drought tolerance rather than heat tolerance has long been a major target worldwide. This thought is in part supported by the fact that South Africa have frequently experienced droughts and literature (e.g., <https://www.grainsa.co.za/screening-bread-wheat-lines-for-drought-tolerance>). However, Table S3 shows that no yield effect of precipitation (and precipitation squared). This suggests that the high temperature effect and drought effect in this study may not be well differentiated each other. To

remedy this concern, I would suggest the author(s) providing Fig. 5 for drought (or precipitation) to show that known drought-tolerant cultivars have lower precipitation effect. Such an additional analysis can increase the reliability and confidence of the findings of this study.

Technical comments:

5. In Table S5, I would appreciate it if the statistical significance of heat effect >30degC for each variety is presented. If the estimated cultivar-specific heat effects that are not significant are included in analysis, it is hard to interpret Fig. 5 b and 5c. I would suggest conducting a separate analysis for all samples and subset with significant heat effect.

6. In Fig. S5, on average, the estimated yield impacts for spring wheat are always more severe than those for winter wheat. Although the difference across the wheat type is not significant (Fig. S5), it would be nice if labels (winter, facultative or spring) are added to Table S2 to enable readers thinking about possible reasons for this non-existence of the significance. As the crop durations are almost the same across the cultivars (Table S2), it is hard to make a distinction (which cultivar is which type?).

Reviewer #2:

Remarks to the Author:

This paper highlights the impacts of extreme heat events on wheat production in South Africa using a large dataset from a number of field plots from a national research center. Interestingly, they also show the potential adaptation benefit of using more advanced seeds. Overall, this paper provides important results that are valuable for scientists and policy makers when it comes to identifying ways to mitigate the impacts of heat on wheat production.

I really liked how thorough the analysis is while reading the methods section, especially the robustness checks. The authors seem to have thought about almost all potential caveats. Well done!

However, I have the following suggestions/concerns/queries:

- The title does not really reflect what you have shown. While very subjective, I wonder whether the title should be more about the potential benefits of plant breeding to mitigate heat impacts? Whatever you go for, I think the paper could use a more convincing title.

- abstract: would be good to end the abstract with a concluding sentence highlighting the importance of this paper.

- introduction: overall, I believe that both the intro and discussion (see below) could use some rethinking/rewriting. There is quite a lot of focus on droughts, which are not the same as heat events. Paragraphs are quite busy. Would be good to include a bit more info on South Africa (who, when and where is wheat produced; import/export; what are the climate models saying about South Africa). From my understanding, there is a strong decrease in the area harvested as the country is more and more relying on imports. Also, it would be good to talk a bit more about irrigation here (you mention it in the discussion). Again, from what I understand, most of the wheat produced in the Free State is irrigated (not so in the Western Cape). And finally, there is no talk at all about soils. Is wheat grown in these two provinces on the same type of soil? different soils are better for keeping moisture. Something that can also be discussed later. (Maybe in the Suppl Mat, you can include some review on the strengths and weaknesses of the various methods that are used to assess the impacts of climate on crops such as Hertel, Thomas W., and Stephanie D. Rosch. Climate change, agriculture and poverty. The World Bank, 2010; or Lobell, David B., and Senthil Asseng. "Comparing estimates of climate change impacts from process-based and statistical crop models." Environmental Research Letters 12.1 (2017): 015001. but also experimental plots).

- discussion: same comments as above, but I would also add that you could further discuss the

caveats (methodological and others), and other issues such as uptake of new seeds. I assume that this mainly focused on large, commercial producers, but I may be wrong. But from my experience with smallholders, there is often a mistrust or other barriers in adapting newer breeds that farmers may not be familiar with. I think it would be interesting to discuss these differences, and what may happen in times of extreme heat events (i suppose smallholders would then mainly rely on markets - if they actually consume wheat (see imports/exports info in introduction)).

-methods

it would be good to include a table or some other detailed summary of what data are available from ARC-SGI. Currently it is rather difficult to understand.

one important concern I have is regarding the use of data from the weather stations, and the somewhat arbitrary buffer of 75km. For temperature, this may be OK (even here there may be big differences depending the landscapes these stations are in) but for rainfall there may be major differences, which also depend on elevation. In Fig 1, one can see that some stations are on the coast while the plots are further inland. Could you maybe compare your rainfall and temperature data with gridded datasets? Or create spatial data, taking into account elevation?

Would it be useful to include other variables such as Growing Degree Days, or Coeff of Variation of Rainfall?

Is there an argument to use these specific, 5C bins? Would the results look different with different bins?

One concern I had from the beginning was spatial autocorrelation. I was glad to read on line 318 that you had also considered this. Not being an expert in this, I would like to know whether your approach is correct. Is there some analysis that can be done to measure existing spatial autocorrelation and highlight the impact of your approach?

The stats are a bit beyond my understanding but I do have two queries: 1. what is the impact of having observations of different length in time? Does this temporal heterogeneity influence results? 2. you do analyse the impact of release year, but what does that really say about the type of cultivar. Are these heat tolerant seeds or do they focus more on specific pests/diseases?

smaller comments:

L202: I believe that ARC-SGI hasn't been defined yet.

L202-213: should this be moved to the discussion?

L286: typo 'iss'

references: there are several where the formatting needs to change (i have identified issues with ref #17 and 48, please check others.)

Figures:

Fig1 - the label/title 'South Africa' is weird as one would think it represents the dark grey areas.

Most figures: not sure why you use color for the barplots, especially fig 3: One would think that the different colors represent something else, and this would then need to be explained somewhere (in caption or legend)

Fig S1: the caption mainly talks about the bottom figure. Provide info on top figure as well. Clearly indicate sub-figures. Label x-axis too.

Fig S2: label both axes

Pedram Rowhani

Reviewer #1 (Remarks to the Author):

General comments:

The submitted manuscript presents a statistical analysis on the high temperature effect on wheat yields in South Africa. The variety-specific data examined here are potentially novel and useful to address the scientific question considered here and lead to implications for breeding targets. The manuscript is mostly clearly written. However, I have several concerns from the methodological point of view, as listed below. Given the current manuscript, rationales are not sufficient or in part lacking to conclude that this study is technically sound; and the conclusions are supported by evidence.

Thank you for your comments. Please see our responses in blue italics below and edits to the manuscript are tracked. We believe the manuscript is much improved based on your contributions.

Major concerns:

1. My main concern is that whether only limited samples in terms of varieties and locations are used to estimate the yield sensitivity to the temperature bin $>30\text{degC}$, although the total sample size is quite large (18,881). Figures S1 and Table S1 suggested that wheat grown in Western Cape is rarely exposed to temperatures $>30\text{degC}$. In Free State, the exposure to the high temperatures may happen for some varieties but not for all varieties as many varieties complete their crop duration by up to the end of September in which the high temperatures rarely occur. The incidences of temperatures $>30\text{degC}$ are shown in Fig. S1, but it is likely that these data came from limited cultivars and locations and probably the sample size is very small. If this is the case, the estimates of yield sensitivity to the high temperatures do not represent the average of various varieties examined here. Subsequently, the comparison across the varieties with low and high sensitivities presented becomes meaningless in this case. The author(s) is strongly encouraged to provide rationale that their estimates on the temperature bin $>30\text{degC}$ is derived based on a wide-ranging varieties and locations.

Table S2 reports the seasonal average of the min and max temperatures, which hides a lot of the day-to-day variation that we actually observe and leverage within the data. This is one of the major strengths of the Schlenker and Roberts (2009) approach that we use to process the weather data into “bins” via interpolation of temperature exposure between the daily min and max. The bottom panel of Figure S1 reports average exposures across all observations for each province, but this again masks a lot of the actual variation inherent in the data.

Nonetheless the concern that credible estimation of the temperature effects requires sufficient exposures across cultivars and locations is a good one, and we thank you for raising it. Within our sample every cultivar has been exposed to temperatures above 30C. For each cultivar, we average across all location-year observations and find that average exposure ranges from 4 to 115 hours. There are however some cultivar-location combinations in which temperatures above 30C are never experienced, but these account for less than 10 percent of total observations. Importantly, even though some cultivars at some locations do not receive 30C exposures, the cultivar itself did experience exposure at other locations in the data.

To further investigate the implications of this for estimating warming impacts, we now consider an additional robustness check where we first discuss the variation in heat exposure across cultivars and cultivar-locations as above, and then re-estimate the model while excluding any cultivar-year that did not experience exposures above 30°C. We find that the warming impacts are very similar (see Figure S8) and thus do not feel that this is a major concern for the analysis. We now include the following text in the methods section for the robustness check:

“It is essential that cultivars in the data experience sufficient heat exposure to capture the temperature effects. Within the sample, every cultivar was exposed to temperatures above 30°C ranging from 4 to 115 hours. Not every cultivar was exposed to temperatures above 30°C at every location, but these account for less than 10 percent of observations. As a robustness check for the warming impact estimates, cultivar-years not experiencing exposures above 30°C were excluded and the model re-estimated. Warming impacts were similar, indicating the effects have likely been captured while including all observations (Supporting Figure S8).”

2. I am not convinced how the findings of this study have implications in particular for Sub-Saharan Africa (SSA), whereas the author(s) sometimes emphasize it in main text (for instance, "... as a major wheat producer and consumer, shocks to South African wheat production via heat extremes likely affect food security outcomes in South Africa and throughout the southern SSA region" (line 48-50)). In reality, South Africa is not necessarily a major wheat producer in the world. According to wheat production and trade data in 2017 available in FAO statistical database, South African wheat production is ranked 5th among African countries following to North and West African countries (e.g., Egypt, Morocco, Ethiopia and Algeria). In 2017, South Africa produced wheat of 1.5 million tons (Mt) and imported 1.7 Mt mainly from Russia and Germany (Table 8 in https://urldefense.proofpoint.com/v2/url?u=https-3A_apps.fas.usda.gov_newgainapi_api_report_downloadreportbyfilename-3Ffilename-3DGrain&d=DwIF-g&c=7ypwAowFJ8v-mw8AB-SdSueVQgSDL4HiiSaLK01W8HA&r=7CKnslRZniKz_Vp4itluxDezTtiKz4tvRoBShnhy18&m=dHDxQpe81vSAlsibwZYdXG4WUqd5PBnV8SvI_LjJNQ&s=vtYaUjnkVEgiOvutAAPROQRbJulkGwGXv_H5YVIALgc&e=<https://urldefense.proofpoint.com/v2/url?u=https-3A_apps.fas.usda.gov_newgainapi_api_report_downloadreportbyfilename-3Ffilename-3DGrain&d=DwMGAg&c=QzRQJIHx0ZTYmlwGx7ptjrPEeuNmnYRxm_FN73lod7w&r=Rla-LO3qiLZnPk78diihLw&m=4opCmyCm5sxKDsDjTferkugaR6unDKT2mp8wMZQAHGM&s=TzfdElt8D9fGwKo67H5jP0fYyFYmEl1rEL9K4RMK1w4&e=>%20and%20Feed%20Annual_Pretoria_South%20Africa%20-%20Republic%20of_3-25-2019.pdf). Wheat exports from South Africa were relatively small (0.08 Mt). Given this situation, the present study may have implications for domestic production in South Africa, but possible implications for the remaining region in SSA as well as those for international scientific communities are unclear.

This is a good point that warrants further explanation. We have revised the terms in the manuscript to be more precise in our meaning. We changed SSA to a more focused regional term “Southern Africa” defined as south of the equator.

We suggest that even while wheat plays a limited role in exports within Southern Africa, South African demand for wheat impacts prices throughout the region because it must/will import what it cannot produce (Dube et al., 2020; Meyer et al., 2016). Thus, when supply within South Africa

decreases due to lack of production, the demand for wheat beyond South African borders increases – likely raising wheat prices regionally. The Regional Network of Agricultural Policy Institutes (ReNAPRI) began providing wheat market outlooks as of 2015 because wheat consumption has been rising for more than a decade and production has remained stagnant. While maize remains the dominant staple, wheat has become important as a supplementary staple food and regardless of imports versus exports, South Africa plays a strong role in mediating the quality and price of wheat within the Southern African context. South African wheat has been bred for quality characteristics, and the wheat that South Africa does export typically ends up mixed with lower quality wheat to improve overall food quality in surrounding countries, including Lesotho, Eswatini, Botswana, and Zimbabwe, among others.

Additionally, South Africa's modern and long-running wheat breeding program likely presents opportunities for higher yields and/or stress resistance in cultivars that may become adopted seed in other SSA countries. In this case, South Africa may not be the largest player in wheat production for food, but may be important in terms of exports of wheat genetics in order for other countries to produce more under challenging conditions. Few wheat breeding programs exist in Southern Africa, so we see this as an important contribution of South Africa.

We have enhanced and edited our discussion around this subject from lines 36-79 in the revised manuscript.

*Dube E, Tsilo TJ, Sosibo NZ, Fanadzo M. Irrigation wheat production constraints and opportunities in South Africa. S Afr J Sci. 2020; 116(1/2), Art #6342, 6 pages.
<https://doi.org/10.17159/sajs.2020/6342>*

*Meyer, F., Traub, L.N., Davids, T., Kirimi, L., Gitau, R., Mpenda, Z., Chisanga, B., Binfield, J., Boulanger, P.; Modelling wheat and sugar markets in Eastern and Southern Africa: Regional Network of Agricultural Policy Research Institutes (ReNAPRI); EUR 28254 EN;
[doi:10.2788/437123](https://doi.org/10.2788/437123)*

3. Sowing date shifts are widely considered as an adaptation measure in crop production. However, this study assumes that the crop durations are unchanged in a warmer climate (+1 to +3 degC). This assumption is unreasonable and unrealistic, and more importantly leads to an overestimation in yield impacts associated with the high temperatures. Therefore, the yield impacts estimates presented need be revised by incorporating adaptation measures that would occur thanks to less additional costs for producers.

Thank you for raising this important point. We expanded the data to include weather variables up to 14 days prior to planting and re-calculated the temperature exposure bins for planting dates shifted to 7 and 14 days earlier respectively. Note, we hold the length-of-season constant such that the cultivar-level planting-to-flowering and flowering-to-harvest days remain fixed, which results in flowering and harvest times that shift by 7 and 14 days, respectively. We use the new planting periods to simulate the warming impacts with the new temperature bins to simulate the potential effects of altering planting dates for +1 to +3°C. The changes can reduce heat impacts only slightly at approximately 1% for +1°C scenarios and by about 4% for +3°C. Here are the models generated:

parm	estimate	stderr	z	p	min95	max95	shift	pup	mature	model
_nl_1	-0.10626	0.036054	-2.9473	0.003206	-0.17692	-0.0356		1	0	30 PreferredU0L30
_nl_1	-0.09797	0.035339	-2.7722	0.005568	-0.16723	-0.0287		1	7	23 PreferredU7L23
_nl_1	-0.08971	0.034888	-2.57145	0.010127	-0.15809	-0.02133		1	14	16 PreferredU14L16
_nl_1	-0.21655	0.062995	-3.43756	0.000587	-0.34002	-0.09308		2	0	30 PreferredU0L30
_nl_1	-0.20046	0.062531	-3.20575	0.001347	-0.32302	-0.0779		2	7	23 PreferredU7L23
_nl_1	-0.18572	0.06238	-2.97721	0.002909	-0.30798	-0.06346		2	14	16 PreferredU14L16
_nl_1	-0.32664	0.081124	-4.02647	5.66E-05	-0.48564	-0.16764		3	0	30 PreferredU0L30
_nl_1	-0.3053	0.081557	-3.74332	0.000182	-0.46514	-0.14545		3	7	23 PreferredU7L23
_nl_1	-0.28675	0.082018	-3.49616	0.000472	-0.4475	-0.12599		3	14	16 PreferredU14L16
note: pup is the planting date shift, 0 is baseline, 7 is one week earlier, 14 is two weeks earlier										
shift is the warming scenario, 1 is +1C etc										
estimate is the warming impact, multiply by 100 to get percent, so -0.1 is a 10% reduction										

We now include Figure S10 in the supporting information and the following in the robustness checks description within the methods section of the manuscript.

“Expanding the weather data also provides an opportunity to consider the potential effects of shifting planting dates. Producers may adapt to increasing heat stress by planting earlier to avoid critical periods of heat exposure. To test the implications of this adaptation, warming impacts were simulated based on the initial temperature impacts with new weather variables created by planting date shifts at 7 and 14 days earlier with fixed (days-to-flowering and days-to-harvest) season lengths. For +1°C, shifting planting dates to 14 days earlier provides approximately one percent reduction in the warming impact on yields, while for +3°C a 14 day earlier planting date may reduce impacts by about four percent (Supporting Figure S10).”

4. Although I am not sure specific targets in wheat breeding in South Africa, drought tolerance rather than heat tolerance has long been a major target worldwide. This thought is in part supported by the fact that South Africa have frequently experienced droughts and literature (e.g.,

This is a great point. Unfortunately, ARC does not include cultivar-level information on breeding purposes, specifically concerning heat/drought stress. ARC breeds for climatic/agronomic regions across South Africa, Free State and Western Cape, respectively, which experience different climatic patterns. Also note that while the separate estimates on precipitation and precipitation squared are not individually statistically significant, they are jointly statistically significant as an F-test for the null hypothesis that both parameters are zero is rejected at conventional significance levels (p -value = 0.0007). In addition, the yield effects of precipitation are not trivial as a one standard deviation reduction in cumulative rainfall below the average level is associated with a 9.6% yield reduction.

To more directly investigate the concern that drought and heat are not well differentiated from each other, we modify the precipitation component to include the quadratic function (as in the preferred model) along with an indicator variable that takes on a value of 1 when cumulative precipitation is below the 10th percentile of all observed rainfall data. This indicator captures low rainfall conditions which are likely associated with droughts, and we find that the effect of drought (defined as 10th percentile of rainfall) is an 18% yield reduction (p -value = 0.003). This seems plausible and is consistent with drought effects in the literature (Daryanto et al. 2016). However, the inclusion of the additional drought control variable does not affect the parameter estimates for the temperature variables in the model as the warming impacts compared to our preferred model are similar. Thus, the high temperature effect and rainfall effect do seem well differentiated from each other. This is likely the case because our preferred model included both location and year fixed effects, which control for (among other things) locations with a more drought-prone climate and widespread droughts across years.

Overall, our focus here is on warming temperatures but you make a great point about model validity and reliability. We now include a discussion on the precipitation effects and the alternative model that includes a drought control in the methods section. The following Figure S11 is also included in the supporting information:

Daryanto S, Wang L, Jacinthe P-A (2016) Global Synthesis of Drought Effects on Maize and Wheat Production. PLoS ONE 11(5): e0156362. <https://doi.org/10.1371/journal.pone.0156362>.

Technical comments:

5. In Table S5, I would appreciate it if the statistical significance of heat effect >30degC for each variety is presented. If the estimated cultivar-specific heat effects that are not significant are included in analysis, it is hard to interpret Fig. 5 b and 5c. I would suggest conducting a separate analysis for all samples and subset with significant heat effect.

Table S5 has been updated with 95% confidence intervals for the estimated heat impacts. These were constructed from block-bootstrapping whole years and re-estimating the model 10,000 times. Of the 47 cultivars, 5 were found to have a CI that contained 0, thereby suggesting a lack of statistical significance. However, the general pattern of results for figure 5 were similar when we excluded these 5 cultivars. Here is the figure with that exclusion:

Since the general pattern of results is similar, we kept figure 5 as is with all 47 cultivars but now mention the similarity when cultivars are excluded in the figure notes due to your good comment. We would be happy to replace the figure entirely with the new one if warranted but feel a bit uneasy about using statistical significance to filter some cultivars out of the analysis.

6. In Fig. S5, on average, the estimated yield impacts for spring wheat are always more severe than those for winter wheat. Although the difference across the wheat type is not significant (Fig. S5), it would be nice if labels (winter, facultative or spring) are added to Table S2 to enable readers thinking about possible reasons for this non-existence of the significance. As the crop durations are almost the same across the cultivars (Table S2), it is hard to make a distinction (which cultivar is which type?).

Great point and recommendation. For clarity, we now include four tables (S2A-D) instead of only Table S2 – A for all sites, B for Facultative, C for Spring, and D for Winter, respectively.

Reviewer #2 (Remarks to the Author):

This paper highlights the impacts of extreme heat events on wheat production in South Africa using a large dataset from a number of field plots from a national research center. Interestingly, they also show the potential adaptation benefit of using more advanced seeds. Overall, this paper provides important results that are valuable for scientists and policy makers when it comes to identifying ways to mitigate the impacts of heat on wheat production.

Thank you for your comments. Please see our responses in blue italics below and edits to the manuscript are tracked. We believe the manuscript is much improved based on your contributions.

I really liked how thorough the analysis is while reading the methods section, especially the robustness checks. The authors seem to have thought about almost all potential caveats. Well done!

Thank you.

However, I have the following suggestions/concerns/queries:

- The title does not really reflect what you have shown. While very subjective, I wonder whether the title should be more about the potential benefits of plant breeding to mitigate heat impacts? Whatever you go for, I think the paper could use a more convincing title.

Great recommendation. We have improved the title accordingly.

- abstract: would be good to end the abstract with a concluding sentence highlighting the importance of this paper.

We added a concluding statement to the abstract as follows: "Findings from this study indicate that warming represents a substantial challenge for wheat producers in South Africa and suggests limited adaptation measures are available to reduce warming impacts."

- introduction: overall, I believe that both the intro and discussion (see below) could use some rethinking/rewriting. There is quite a lot of focus on droughts, which are not the same as heat events. Paragraphs are quite busy. Would be good to include a bit more info on South Africa (who, when and where is wheat produced; import/export; what are the climate models saying about South Africa). From my understanding, there is a string decrease in the area harvested as the country is more and more relying on imports. Also, it would be good to talk a bit more about irrigation here (you mention it in the discussion). Again, from what I understand, most of the wheat produced in the Free State is irrigated (not so in the Western Cape).

Good points. We have revised the introduction of the manuscript according to your recommendations. The revisions should be visible in track changes.

Historically, South Africa has been the second largest wheat producer (by area and production) in Sub-Saharan Africa behind Ethiopia. In 1998 wheat acreage declined by 46% due to the

deregulation of the wheat market and abolishment of the fixed pricing system by the wheat marketing board. Since 1998, South Africa has been both an importer (typically of lower quality wheat to blend with high quality domestic wheat) and an exporter (predominately to other Southern African Development Community members). The drought of 2015-2016 saw South African wheat exports drop by over 76% (FAOSTAT). The drought hit the Western Cape the hardest as it is South Africa's largest wheat growing province and is dominated (>90%) by dryland production. Free State, the second largest wheat producing province, is a mix of dryland and irrigated production (where dryland wheat is the predominant method of production but irrigated wheat accounts for more than 50% of the total yield). The relatively low productivity of wheat across Sub-Saharan Africa is principally because of abiotic (drought and heat) and biotic (yellow rust, stem rust, septoria and fusarium) stresses, which are increasing in intensity and frequency under climate change (Tadesse et al. 2019). Previous literature has suggested that the estimated drier conditions in South Africa could reduce wheat yields from 1.8% to 4.3% annually (Cullis et al. 2015). Further, because of the anticipated drier conditions it is predicted that irrigation usage in South Africa will increase 6.4% a year through 2050 stressing the already limited water availability even further.

Cullis, J. et al. An uncertainty approach to modelling climate change risk in South Africa. vol. 2015 (UNU-WIDER, 2015).

Tadesse, W., Bishaw, Z. & Assefa, S. Wheat production and breeding in Sub-Saharan Africa: Challenges and opportunities in the face of climate change. International Journal of Climate Change Strategies and Management 11, 696–715 (2019).

And finally, there is no talk at all about soils. Is wheat grown in these two provinces on the same type of soil? different soils are better for keeping moisture. Something that can also be discussed later.

We agree that soils could play a major role in moisture retention, but unfortunately we do not have data on the soils at trial sites. We do, however, control for all time-invariant factors using site fixed effects, which will likely account for features such as soil moisture retention. We also include a robustness check utilizing a proxy for soil moisture; specifically, we interact low precipitation with the high temperature bins. This indicator captures low rainfall conditions likely associated with droughts, and findings suggest the effect of 10th percentile rainfall is an 18% yield reduction (p-value = 0.003). The inclusion of the additional drought control variable did not affect the parameter estimates for the temperature variables in the model as the warming impacts compared to the preferred model are similar (Supporting Figure S11). Thus, the high temperature effect and precipitation effect seem well differentiated from each other, in part due to location and year fixed effects that control for (among other things) locations with a more drought-prone climate and widespread droughts across years. See the additional robustness check added for this at line ~455.

(Maybe in the Suppl Mat, you can include some review on the strengths and weaknesses of the various methods that are used to assess the impacts of climate on crops such as Hertel, Thomas W., and Stephanie D. Rosch. Climate change, agriculture and poverty. The World Bank, 2010; or Lobell, David B., and Senthold Asseng. "Comparing estimates of climate change impacts from

process-based and statistical crop models." Environmental Research Letters 12.1 (2017): 015001. but also experimental plots).

There is a brief discussion about the modelling approach and potential for synergistic analysis with process-based, biophysical models. We added a more explicit suggestion for readers to look at the Lobell and Asseng (2017) and/or Roberts et al. (2017) papers for more detailed explanations. We are hesitant to assume/compare how biophysical models compare with our empirical models in this specific study, and we feel that these other articles discuss the general issues, benefits, and trade-offs associated with different approaches. The following was added at line ~275.

"There are trade-offs between empirical estimations as in this study and biophysical crop models. For an extensive discussion of this topics, see Lobell and Asseng (2017) and Roberts et al. (2017)."

- discussion: same comments as above, but I would also add that you could further discuss the caveats (methodological and others), and other issues such as uptake of new seeds. I assume that this mainly focused on large, commercial producers, but I may be wrong. But from my experience with smallholders, there is often a mistrust or other barriers in adapting newer breeds that farmers may not be familiar with. I think it would be interesting to discuss these differences, and what may happen in times of extreme heat events (i suppose smallholders would then mainly rely on markets - if they actually consume wheat (see imports/exports info in introduction)).

This is a good point. We now highlight that this is primarily focused on commercial producers and provide a brief statement of potential implications for small-holders in the discussion. Unlike maize where subsistence farming is a major driver in food security and overall production, wheat is much more commercialized with very little production by small-scale or subsistence farming. With the spirit of your comment in mind, there does not seem to be a barrier to adoption between private (Sensako and Pannar who have well over 50% of the market) and public (ARC) seed varieties amongst commercial farmers, although there is a price premium for private lines.

-methods

it would be good to include a table or some other detailed summary of what data are available from ARC-SGI. Currently it is rather difficult to understand.

Thank you for point this out. We state this more explicitly in the methods section now, and reference the supplementary tables where information can be found in Supp Info Table 1 and 2A-D. The table footnotes cite which data are from ARC-SGI, which are calculated, and which are from NASA GSOD.

one important concern I have is regarding the use of data from the weather stations, and the somewhat arbitrary buffer of 75km. For temperature, this may be OK (even here there may be big differences depending the landscapes these stations are in) but for rainfall there may be major differences, which also depend on elevation. In Fig 1, one can see that some stations are on the

coast while the plots are further inland. Could you maybe compare your rainfall and temperature data with gridded datasets? Or create spatial data, taking into account elevation? Compare elevation of weather station and elevation of site

This is a very good point, and we struggled with the correct buffer distance. Most gridded data such as WorldClim2 are interpolated using the same/similar weather station data, and gridded data are often limited to monthly observations on a 0.5x0.5 degree grid – not much different/better than what we have. Additionally, when weather data are interpolated to a grid, we tend to lose the more extreme observations that may have a significant impact on crops. Thus there's a trade-off between the potentially more consistent/accurate gridded data and the potential extremes that may exist at the site locations.

In due diligence, we interpolated weather data using all stations within 200 km of site locations. The interpolation was constructed similar to creating a gridded dataset as a distance-weighted (weight=1/dist²) daily measure of tmax, tmin, and precip. Temperature bins were re-constructed from the interpolated data and re-estimated the models. See the yield impacts using the interpolated data below. Our results were robust to interpolated spatial weather information, and results were not significantly different from our preferred model using the simpler closest weather station data within 75 km. This is included as a robustness check in the methods section at line ~401.

Would it be useful to include other variables such as Growing Degree Days, or Coeff of Variation of Rainfall?

Growing Degree Days are typically used to identify the minimum temperature required for a crop to grow and is cumulative through maturity. In this case, we prefer to use temperature bins so as to capture not only the effects of extreme heat (>30C), but also control for the potential beneficial effects provided between 20C and 30C. This can be observed in Schlenker & Roberts PNAS 2009 article (<https://www.pnas.org/content/106/37/15594>), although wheat is not analysed specifically in their study. In that study the “piecewise linear” approach that leverages degree days is shown to be a restricted version of the more general “exposure bin” approach that we use here, so in that sense our modelling of the nonlinear effect of temperatures is more flexible than ones based on degree days.

We prefer to use annual precipitation outcomes in the regression model as our goal is to link annual yield and weather outcomes. We think of the coefficient of variation as a climate (not weather) variable since it’s construction requires averaging over outcomes to measure the first two moments of the distribution (here we are thinking of the CV as sigma/mu). Averaging smooths out variation that we would prefer to leverage. However, we would be happy to reconsider if this argument is not convincing.

Is there an argument to use these specific, 5C bins? Would the results look different with different bins?

Our main interest is in the extreme temperatures that affect wheat, hence the focus of our manuscript on the >30C bin. The results could change slightly for bins below 30C, but our focus is on warming scenario impacts. We use 30C (1) because it is a documented heat threshold for wheat (Barlow et al. 2015, Liu et al. 2013, Semenov and Shewry 2011) and (2) because raising the upper threshold from 30C to something higher would reduce the amount of heat exposure and variation in heat exposure needed to estimate the warming impacts. Supplementary Figure S1 and S2 illustrate the variability of temperatures within bins; note the amount of exposure greater than 30C. We added a sentence along with these citations to the methods sections discussing temperature bins.

“The highest temperature bin of >30°C represents exposures known to potentially affect wheat yields⁴⁹⁻⁵¹.”

*Barlow, K. M., Christy, B. P., O’Leary, G. J., Riffkin, P. A. & Nuttall, J. G. Simulating the impact of extreme heat and frost events on wheat crop production: A review. *Field Crops Research* 171, 109–119 (2015).*

*Liu, B. et al. Post-heading heat stress and yield impact in winter wheat of China. *Global Change Biology* 20, 372–381 (2014).*

*Semenov, M., Shewry, P. Modelling predicts that heat stress, not drought, will increase vulnerability of wheat in Europe. *Sci Rep* 1, 66 (2011).*

One concern I had from the beginning was spatial autocorrelation. I was glad to read on line 318 that you had also considered this. Not being an expert in this, I would like to know whether your

approach is correct. Is there some analysis that can be done to measure existing spatial autocorrelation and highlight the impact of your approach?

Thank you for the comment. We are accounting for spatial correlation using clustered standard errors, but have not done a good job of communicating the strengths of this approach. Typically the use of such errors is motivated by the suspicion that the error terms from the regression model are spatially correlated, which is fair, but the method also accounts for correlations among the regressors which can also bias standard errors if not accounted for.

Cameron and Miller (2015) report the variance inflation factor in eq 6 as:

$$1 + \rho_x \rho_u (N - 1),$$

where N is the cluster size, ρ_u is the within-cluster correlation of the regression errors, and ρ_x is the within-cluster correlation of the regressor. Note that spatial correlation of the regressors (which is almost surely the case with temperature and to a lesser extent precipitation) can bias regression standard errors downward even if the errors are only slightly correlated. Just under their equation 6, Cameron and Miller (2015) cite a study in which the correlation of the errors was small at 0.03 but the inflation factor was 13 because the regressors were highly correlated.

To better investigate the role that spatial correlation is playing in our analysis, we calculated Moran's I for each year in the data for both the raw yield data and the regression errors from our preferred model. Here are the averages of the Moran's I across years. The distance of 1 km captures the within-trial correlations, whereas the distances 100, 500, and 1000 km capture broader groupings.

data	MI1	MI100	MI500	MI1000
raw	0.34404	0.319946	0.157958	0.097185
errors	0.178149	0.138631	0.058130	0.026231

We see that there exists positive correlation in the raw data and it is highest within-trial as we would expect. The correlation remains positive as we move further out but does begin to dilute to its smallest value at 1000 km. The regression purges much of the correlation from the data as indicated by the Moran's I for the errors, but some remains.

As noted above the clustered errors can still produce a big adjustment in the variances. We find this to be the case as the clustered standard errors are much larger than the "classical OLS" ones that do not account for correlations. For example, the standard error on our measure of heat (the 30+C bin) is 0.0196 under clustering but .00433 without clustering (ie "robust" standard errors). This suggests an inflation factor of approximately 20 which is quite large and important for adequately representing the statistical uncertainty in our warming impacts.

Thank you for the opportunity to clarify this important point.

Cameron, A. Colin, and Douglas L. Miller. "A practitioner's guide to cluster-robust inference." Journal of human resources 50.2 (2015): 317-372.

The stats are a bit beyond my understanding but I do have two queries:

1. what is the impact of having observations of different length in time? Does this temporal heterogeneity influence results?

The estimated models are by cultivar based on growing seasons in each year and site so the estimated effects are specific to the temperature exposures for each observed growing season. Schlenker & Roberts (2009) referenced above include a robustness check for time and specified growing seasons for these procedures across a number of functional forms and find no significant difference. You can view their detailed analysis of in the supplementary information (section 10) of their article here:

https://www.pnas.org/content/suppl/2009/08/26/0906865106.DCSupplemental/Appendix_PDF.pdf.

2. you do analyse the impact of release year, but what does that really say about the type of cultivar. Are these heat tolerant seeds or do they focus more on specific pests/diseases?

Good point. Unfortunately, ARC does not provide specific information on breeding purposes for cultivars, specifically as it pertains to heat/drought stress. There are ratings (albeit subjective) about disease and pest resistance but none for drought/heat stress resistance. Anecdotally, we know that cultivars tend to be bred for Free State or Western Cape, respectively, but other than the statistical significance and size of impacts by cultivar in Table S5, we are hesitant to make further inferences about the cultivars from a purpose standpoint.

The purpose of using release year is to capture potential advances in heat resilience through time. In other words, it tests the hypothesis that breeding has led to varieties with improved yield and/or increased heat resilience. This uses release year as a proxy for the breeding progress with time.

The only other cultivar "type" we are able to test with the data we have is for facultative, winter, and spring, which we now include summary statistics for in Tables S2B-S2D. These tables along with the model presented in Fig. S5 suggest no significant heat impact difference between types.

smaller comments:

L202: I believe that ARC-SGI hasn't been defined yet.

Fixed.

L202-213: should this be moved to the discussion?

Good suggestion. We moved this to the first paragraph of discussion.

L286: typo 'iss'

Fixed.

references: there are several where the formatting needs to change (i have identified issues with ref #17 and 48, please check others.)

Fixed.

Figures:

Fig1 - the label/title 'South Africa' is weird as one would think it represents the dark grey areas.

Fixed.

Most figures: not sure why you use color for the barplots, especially fig 3: One would think that the different colors represent something else, and this would then need to be explained somewhere (in caption or legend)

Good point. We changed Figure 3 to a single color bar plot. Other barplots use different colors to represent secondary categories highlighted in legends.

Fig S1: the caption mainly talks about the bottom figure. Provide info on top figure as well. Clearly indicate sub-figures. Label x-axis too.

Fixed.

Fig S2: label both axes

Fixed.

Reviewers' Comments:

Reviewer #1:

Remarks to the Author:

The author(s) satisfactorily addressed my concerns. I only suggest some minor edits. Although I believe these edits would improve clarity and readability of the manuscript, the decision on adoption or rejection of them is entirely left for the author(s).

1. L9. "a 12.5% yield reduction". I think, the confidence interval needs be mentioned in Abstract.
2. L9-10. "The yield models are then used ...". Please consider presenting your findings in Abstract, but not what you did. What is the estimated yield reduction for +1 to +3 degC? Their confidence intervals are also required.
3. L11-13. The same suggestion is applied here. I think, a main implication of this study which is worth mentioning in Abstract is described around L243-245.
4. L27. "aggregate, non-empirical". Do you mean "biophysical" crop model?
5. L85. "extreme weather". "The drought event" would be more precise if the extreme weather indicate drought.
6. L130. Please consider adding the confidence interval for the estimates.
7. L212-214. I would suggest citing Fig. S3 here to show the rationale.
8. L226. "-10 and -11 percent". Are these relative to the multi-variety average? If so, I would ask the author(s) explicitly mentioning the baseline.
9. L274. There is a variation in terminology (biophysical crop model, crop growth model etc.). Please use a consistent terminology.

Reviewer #2:

Remarks to the Author:

The manuscript has now substantially improved following the authors' response to the feedback received from all reviewers. However, I still have some questions as some of the responses fall short (and there may be some new ones).

- There is still a lot of talk about droughts in the introduction. The manuscript deals with warming/heat so it shouldn't use drought examples and papers as support.

- There is a lot of similarities with Schlenker and Roberts (2009), which is great. But they focused (as far as I understood it) on 1C bins. As I asked previously, what would happen if you chose other bin sizes? Why is the threshold set at 30C and not 29C or 31C? Also, the references to support the threshold of 30C focus on other regions, and possibly other cultivars, and I assume this threshold could be different here. I'd like to see some sensitivity analysis.

- precipitation data: a few things here. first, I do not see that robustness check that you mention 'at line ~401'. second, there are fairly good daily datasets (TAMSAT and CHIRPS) at higher resolution than WorldClim2. third, should you interpolate, then you need to use a method that takes into account elevation and other factors (like ANUSPLIN). Or you could use the weather stations to bias correct the gridded data over the stations. fourth, the distribution of rainfall within a season is also important, in addition to the seasonal sum. I was suggesting to use the seasonal CV (see Rowhani et al. 2011) but if you can think of another way, that would also be great. and finally, you include low rainfall to the highest temperature bin only. What would happen if that interaction was included with all the bins?

- could all the robustness check also show a similar figure to Fig 2? just like in Schlenker and Roberts.

- climate change impacts: the cited papers (L108) refer mainly to studies in the US, what do the climate models say for South Africa? It would make more sense using those numbers than relying on other regions (and other studies). And how do you include changes in rainfall patterns when looking at the impacts of climate change?

- L58-60: I suppose you mean 'production' and not 'productivity'. Average yield seems to be around 4 tonnes/ha which is quite similar to other areas that produce rainfed wheat (eg., <https://www.frontiersin.org/articles/10.3389/fpls.2015.00990/full>). Overall productivity usually has to do with management, seeds, soils,... and can be affected by these biotic and abiotic factors that you mention during an event.

- Timing is always an important factor, especially with heat events. Is there a way to check the model output when looking at temperature profiles for specific phases of plant growth (i.e., from planting to flowering and flowering to harvest)?

- out of curiosity, and as a result from the spatial autocorrelation analysis, how confident can one be with your results given the high inflation factor? also, given the differences in climate and other factors between the two provinces, would it make sense to run two separate models (one for each province) or have province as a fixed effect (or a hierarchical fixed effect site:province - but I am way out of my expertise).

- finally, it is a shame you do not have more info on the cultivars because that is really the main point of the paper and it would allow us to properly understand what is going on. While I understand that there is not much you can do about the data itself, I am wondering whether you can provide some more information in some other way. For example, in Fig 5, are there cultivars that have low heat impact and high yield (maybe I am thinking of all of those way above the red line in Fig5C). Could you look at yield trends for these? Since this is a paper on breeding as adaptation, it would be good to provide more info one way or another.

- Table 1 needs reformatting

- l25: could use some citations

- references 15 and 19:remove the letter after 'GAIN'

REVIEWER COMMENTS

Reviewer #1 (Remarks to the Author):

The author(s) satisfactorily addressed my concerns. I only suggest some minor edits. Although I believe these edits would improve clarity and readability of the manuscript, the decision on adoption or rejection of them is entirely left for the author(s).

Thank you for the additional editorial comments. We have changed the manuscript accordingly.

1. L9. "a 12.5% yield reduction". I think, the confidence interval needs be mentioned in Abstract.

Done

2. L9-10. "The yield models are then used ...". Please consider presenting your findings in Abstract, but not what you did. What is the estimated yield reduction for +1 to +3 degC? Their confidence intervals are also required.

Done

3. L11-13. The same suggestion is applied here. I think, a main implication of this study which is worth mentioning in Abstract is described around L243-245.

Done

4. L27. "aggregate, non-empirical". Do you mean "biophysical" crop model?

Done

5. L85. "extreme weather". "The drought event" would be more precise if the extreme weather indicate drought.

Done

6. L130. Please consider adding the confidence interval for the estimates.

Done

7. L212-214. I would suggest citing Fig. S3 here to show the rationale.

Done

8. L226. "-10 and -11 percent". Are these relative to the multi-variety average? If so, I would ask the author(s) explicitly mentioning the baseline.

Done

9. L274. There is a variation in terminology (biophysical crop model, crop growth model etc.). Please use a consistent terminology.

Done

Reviewer #2 (Remarks to the Author):

The manuscript has now substantially improved following the authors' response to the feedback received from all reviewers. However, I still have some questions as some of the responses fall short (and there may be some new ones).

Thank you once again for the critiques and comments. We have addressed your concerns in the manuscript and point-by-point below.

- There is still a lot of talk about droughts in the introduction. The manuscript deals with warming/heat so it shouldn't use drought examples and papers as support.

We have removed most of the drought mentions with exception to the recent impacts of drought on food security and the South African wheat economy. We feel that it is important to discuss weather in general because it sheds light on how weather shocks – regardless of their source as heat or drought – can impact regional markets and human well-being by reducing yields and production. Heat impacts on wheat have been studied less than drought impacts in South Africa. We change our verbiage to highlight that substantially more work has been done on drought impacts on wheat in the South African context, which we believe highlights the importance of our study. We try to hone the rhetoric in this direction and state in the final introductory paragraph what the aim and purpose of our study is – establishing the impacts of temperature exposure on wheat yields. Additional language is provided on L98-113.

- There is a lot of similarities with Schlenker and Roberts (2009), which is great. But they focused (as far as I understood it) on 1C bins. As I asked previously, what would happen if you chose other bin sizes? Why is the threshold set at 30C and not 29C or 31C? Also, the references to support the threshold of 30C focus on other regions, and possibly other cultivars, and I assume this threshold could be different here. I'd like to see some sensitivity analysis.

Thank you very much for the suggestions and we apologize for not addressing them during the previous review. We now have addressed these concerns as follows. First, we estimate a 3C bin model as in Schlenker and Roberts (2009). Note that in that paper they do construct the exposures using 1C bins, but when they estimate the model they aggregate them into 3C bins. We take the same approach here and find that it produces similar warming impacts as the 5C bin model. We also investigate the upper threshold for values of 29C and 31C and again find similar impacts. The results for these alternatives have been added to the “Robustness Checks” section on pg 22. This robustness check is described starting at L424; also see Supporting Figures S5 and S6.

- precipitation data: a few things here. first, I do not see that robustness check that you mention 'at line ~401'. second, there are fairly good daily datasets (TAMSAT and CHIRPS) at higher resolution than WorldClim2. third, should you interpolate, then you need to use a method that takes into account elevation and other factors (like ANUSPLIN). Or you could use the weather stations to bias correct the gridded data over the stations. fourth, the distribution of rainfall within a season is also important, in addition to the seasonal sum. I was suggesting to use the seasonal CV (see Rowhani et al. 2011) but if you can think of another way, that would also be great. and finally, you include low rainfall to the highest temperature bin only. What would happen if that interaction was included with all the bins?

Thank you for the suggestions, we discuss each of them in turn here.

As noted we previously employed a robustness check using interpolations across all weather stations, for which a concern on the interpolation technique was noted. The suggestion to employ higher resolution data is made here, and we believe that this also alleviates the interpolation concern since the new data that we employ has already been downscaled to the grid level using (potentially more credible) interpolation techniques. To this end, we obtained the daily CHIRPS precipitation data and used it in the regression model instead of the precipitation data that we had interpolated ourselves. We find that all three approaches, (i) the initial use of only weather stations with a full time series of data available matched to trial sites (i.e., the “preferred approach” in the paper), (ii) the interpolation across all available weather stations regardless of time series dimension; and (iii) using instead the precipitation measure from the CHIRPS data all generated similar warming impacts. The results for these alternatives have been added to the “Robustness Checks” section on pg 23, L429; also see Supporting Figures S16 and S17.

The next comment suggested using the seasonal CV of precipitation from the Rowhani et al paper. For each site-year-growing season observation, we calculated both the mean and standard deviation of precipitation across days. Then we formed the CV as the ratio of the standard deviation over the mean. That paper was a little unclear on how exactly the measure was constructed from the daily data so we are happy to measure it differently if this is not correct. We then include the CV as an additional control variable in the model and find similar warming impacts. The results have been added to the “Robustness Checks” section on pg 23, L440; also see Supporting Figures S8 and S9.

The final comment suggests including interactions between low precip and all the temp bins. We find that this extension did not improve model performance as the root mean squared error from the out of sample prediction exercise is 0.31 for both (i) the model that interacts low precip with only the high heat bin, and (ii) the model that interacts it with all temp bins. So either these additional interactions are not warranted or the data does not contain sufficient variation to disentangle all of these interaction effects. We did run the warming impacts for the full interaction model and the warming impacts for 1, 2, and 3C are yield increases of 33, 87, 157%. These impacts are nonsensical as hotter temperatures in low rainfall years should not increase crop yields, especially by such large magnitudes. We take this as evidence that there is not sufficient variation in the data to estimate such a wide range of tempXprec interaction effects. For these reasons we do not include these additional results in the manuscript, but would be happy to revisit this if warranted.

- could all the robustness check also show a similar figure to Fig 2? just like in Schlenker and Roberts.

Thank for you the suggestion. We have added several additional robustness checks to the manuscript, so we took this opportunity to re-organize that section and provide additional figures and supplementary information to accompany it. We did not remove any results from that section, we added new material and changed some verbiage to make it all flow together. As before we include warming impacts for the alternative models. Following the suggestion here, we added figures for the marginal effects of temperatures as well, which can be seen in supplementary figures S6, S9, S13, S17.

- climate change impacts: the cited papers (L108) refer mainly to studies in the US, what do the climate models say for South Africa? It would make more sense using those numbers than relying on other regions (and other studies). And how do you include changes in rainfall patterns when looking at the impacts of climate change?

Good point. We have added more citations and statements for climate change in South Africa and highlight the regional climate change expectations as suggested in a new paragraph. The references used at previous L108 were highlighting other empirical studies of warming impacts on wheat using similar methods, albeit in other regions because so few studies have been conducted on heat and wheat beyond biophysical crop models.

We added a new paragraph (L98-113) specifically discussing IPCC and other climate change projections for Southern Africa, and highlight that land surface temperatures are expected to be higher regionally compared with the projected global temperature increases. While the climate models tend to project less rainfall for the region, there remains substantial uncertainty on this and studies have found differing signs for precipitation (increases and decreases). Due to the uncertainties surrounding future rainfall, we use current rainfall as a control variable given our primary focus is on estimating warming impacts under current conditions as well as three future uniform warming scenarios.

- L58-60: I suppose you mean 'production' and not 'productivity'. Average yield seems to be around 4 tonnes/ha which is quite similar to other areas that produce rainfed wheat (eg.,

soils,... and can be affected by these biotic and abiotic factors that you mention during an event.

Changed accordingly.

- Timing is always an important factor, especially with heat events. Is there a way to check the model output when looking at temperature profiles for specific phases of plant growth (i.e., from planting to flowering and flowering to harvest)?

Thanks for the suggestion. We investigated this by separating the growing season into three stages: (i) planting to 20 days before flowering to capture the vegetative stage, (ii) 20 days before to 10 days after the flowering date to capture the flowering stage, and (iii) 10 days after flowering to the end of season to capture the grain-filling stage. We consider these three stages instead of the two suggested in the comment because flowering is typically considered a separate physiological stage distinct from vegetative and grain fill. We re-estimate the model allowing the effect of all weather variables to differ across each stage and then aggregate (across the three stages) warming impacts into total yield loss. We find that the impacts are very similar to those from our current approach. The results have been added to the “Robustness Checks” section on pg 24, L445-451.

- out of curiosity, and as a result from the spatial autocorrelation analysis, how confident can one be with your results given the high inflation factor? also, given the differences in climate and other factors between the two provinces, would it make sense to run two separate models (one for each province) or have province as a fixed effect (or a hierarchical fixed effect site:province - but I am way out of my expertise).

Great question. Here is how we think of the general issue.

Typically (we are very broadly stereotyping here) statisticians want to explain as much of the variation in the data as possible and “clean up” sources of autocorrelation directly, while econometricians want to get a credible estimate of the parameter(s) of interest and then be as cautious as possible with the statistical inference by accounting for multiple sources of deviations from the classical i.i.d regression model assumption for the errors. [Maybe it’s more appropriate to just say “two camps” as the two disciplines have had increased overlap over the last few decades and are increasingly learning from each other.] We are in the latter camp here.

*We think we have gotten fairly reliable estimates of the temperature and warming effects (backed up by the extensive robustness checks), but do not assume that we have cleaned up all the correlations within the errors. This would take an extensive amount of additional work. So we are trying to be cautious in the statistical inference by allowing for both heterogeneity in the variance of the errors (*robust* standard errors) and spatial correlations within the cross-section (*cluster-robust* standard errors). It’s entirely possible that allowing for these generalities has led to higher standard errors than what is *correct*, but we have decided to error on the side of caution.*

Our view is that the regression model we utilize is in-line with hierarchical models in that some amount of partial pooling is being done to leverage information. The intercept is being allowed

to vary at the location-year-cultivar level; however, the temperature effects are assumed to be the same across them. We devote much attention to potential violations of this latter assumption by allowing for some of the weather effects to vary across certain dimensions and exploring the implications. It remains an open question as to whether we have not considered enough of the possible dimensions, but we feel that is for future research to consider at this point.

Under our modelling approach we include location fixed effects. These will naturally embed any fixed effects at higher levels of spatial aggregation (i.e. province level). In addition, we tested whether the precipitation and heat effects were different across provinces by interacting a province dummy variable with them. Neither interaction was individually statistically significant ($p > 0.15$ for both), nor jointly ($p > 0.30$). So we did not pursue this further.

- finally, it is a shame you do not have more info on the cultivars because that is really the main point of the paper and it would allow us to properly understand what is going on. While I understand that there is not much you can do about the data itself, I am wondering whether you can provide some more information in some other way. For example, in Fig 5, are there cultivars that have low heat impact and high yield (maybe I am thinking of all of those way above the red line in Fig5C). Could you look at yield trends for these? Since this is a paper on breeding as adaptation, it would be good to provide more info one way or another.

Great suggestion. Note that while we do not have more information on the cultivars, we do provide detailed results for each of them alongside their name. Researchers can focus on a set of cultivars for which they find the results interesting and use the name to procure more information, be it genetic and/or phenotypic. To more specifically address this comment, we now replicate figure 5 in Supporting Figure S3 for the 10 cultivars with the highest heat ratio (heat effect over mean yield in 5C). We see that in general these cultivars are following the same pattern as the general sample: increasing mean yields, decreasing heat resilience, increasing ratio of heat/mean. Our take away is that this seems to be a broad trend in breeding efforts within this sample. We briefly mention this result and additional figure on pg 12, L203-204 in the manuscript.

- Table 1 needs reformatting

Done

- 125: could use some citations

Done

- references 15 and 19:remove the letter after 'GAIN'

Changed accordingly.